# From the Lab to the Field: Long-Distance Transport of Sterile *Aedes* Mosquitoes

**DOI:** 10.3390/insects14020207

**Published:** 2023-02-18

**Authors:** Hamidou Maïga, Mame Thierno Bakhoum, Wadaka Mamai, Gorgui Diouf, Nanwintoum Séverin Bimbilé Somda, Thomas Wallner, Claudia Martina, Simran Singh Kotla, Odet Bueno Masso, Hanano Yamada, Bazoumana B. D. Sow, Assane Gueye Fall, Jeremy Bouyer

**Affiliations:** 1Insect Pest Control Laboratory, Joint FAO/IAEA Centre of Nuclear Techniques in Food and Agriculture, Department of Nuclear Sciences and Applications, IAEA Laboratories, Seibersdorf, P.O. Box 100, A-1400 Vienna, Austria; 2Institut de Recherche en Sciences de la Santé/Direction Régionale de l’Ouest (IRSS-DRO), Bobo-Dioulasso BP 2779, Burkina Faso; 3Laboratoire National de l’Elevage et de Recherches Vétérinaires, Institut Sénégalais de Recherches Agricoles (ISRA), Dakar BP 2057, Senegal; 4Institut de Recherche Agricole pour le Développement (IRAD), Yaoundé P.O. Box 2123, Cameroon; 5Unité de Formation et de Recherche en Sciences et Technologies (UFR/ST), Université Norbert ZONGO (UNZ), Koudougou BP 376, Burkina Faso

**Keywords:** mass rearing, sterile insect technique, *Aedes aegypti*, shipment, SIT

## Abstract

**Simple Summary:**

Pilot sites of the sterile insect technique (SIT) may not be close to the rearing facility and so the outsourcing of sterile males may be needed. This study, therefore, aimed to develop and assess a novel method for long-distance shipments of sterile male mosquitoes from the laboratory to the field. In addition, a simulated transport of marked and unmarked sterile males was assessed in terms of survival rates/recovery rates, flight ability and damage to the mosquitoes. The novel mass transport protocol allowed a long-distance shipment of 50,000 sterile male mosquitoes for up to four days without a significant impact on the above-mentioned parameters. In addition, a one-day recovery period for transported mosquitoes post-transport increased their ability to fly. This novel system for long-distance mass transport of mosquitoes may therefore be used to ship sterile males worldwide for journeys of two to four days.

**Abstract:**

Pilot programs of the sterile insect technique (SIT) against *Aedes aegypti* may rely on importing significant and consistent numbers of high-quality sterile males from a distant mass rearing factory. As such, long-distance mass transport of sterile males may contribute to meet this requirement if their survival and quality are not compromised. This study therefore aimed to develop and assess a novel method for long-distance shipments of sterile male mosquitoes from the laboratory to the field. Different types of mosquito compaction boxes in addition to a simulation of the transport of marked and unmarked sterile males were assessed in terms of survival rates/recovery rates, flight ability and morphological damage to the mosquitoes. The novel mass transport protocol allowed long-distance shipments of sterile male mosquitoes for up to four days with a nonsignificant impact on survival (>90% for 48 h of transport and between 50 and 70% for 96 h depending on the type of mosquito compaction box), flight ability, and damage. In addition, a one-day recovery period for transported mosquitoes post-transport increased the escaping ability of sterile males by more than 20%. This novel system for the long-distance mass transport of mosquitoes may therefore be used to ship sterile males worldwide for journeys of two to four days. This study demonstrated that the protocol can be used for the standard mass transport of marked or unmarked chilled *Aedes* mosquitoes required for the SIT or other related genetic control programs.

## 1. Introduction

Mosquitoes are major vectors of human disease pathogens worldwide. *Aedes aegypti* (Linnaeus) is highly invasive [1] and transmits several arboviruses and related diseases such as dengue, chikungunya, yellow fever and Zika [2,3].

Sole reliance on source removal and insecticides is limited in scope and success [4] and therefore alternatives and/or complementary tools are needed [5]. In addition, a limited number of effective vaccines or drugs is available to protect against the aforementioned diseases. Fortunately, there is renewed interest in genetic control strategies, including the sterile insect technique (SIT). The SIT is based on repeated releases of sterile insects to induce sterility in the wild population, thereby suppressing the target pest species [6]. Over the last two decades, the Insect Pest Control Laboratory (IPCL) of the Joint Food and Agriculture Organization/International Atomic Energy Agency (FAO/IAEA) Centre of Nuclear Techniques in Food and Agriculture has invested great efforts in the SIT in field projects in several Member States, which has led to positive results (reviewed in [7]).

Several aspects of the SIT package have been studied, developed and refined—including the details of how best to mass rear, irradiate, mark, handle, release and monitor the mosquitoes [7]—but there are some areas where further study may greatly improve the operational success of such programs. Transboundary shipments of mosquitoes by air may take more than 24 h [8,9] and recent studies have shown contrasting results for mosquitoes transported as pupae or as chilled or nonchilled sterile males. Furthermore, in these studies, the results were relevant for transportation lasting no more than 24 h [9,10,11,12,13,14]. In contrast, operational fruit fly SIT programs entail the long-distance transportation of pupae under different atmospheric conditions (for 67 h to 89 h) without impacting the quality of the flies [15,16]. A recent document on the International Guideline for Transboundary Shipments of Irradiated Sterile Insects was published by the FAO/IAEA to help guide the formulation of a more appropriate and harmonized regulatory framework for safe and timely transboundary shipments of irradiated sterile insects for SIT development and applications [8].

Mosquitoes need to be chilled and compacted during the flight to reduce physical damage to the mosquitoes being transported. If drones are used [17], once the sterile males are delivered, they should be transferred into a drone release device. In some programs where ground release is the aim, chilled mosquitoes should be maintained in single cells with a density that directly transfers to the corresponding release cages/cups, with a minimum recovery time for mosquitoes to regain movement and activity. Sterile males are marked before release in the field for most of the SIT pilot programs [17]. Marking using fluorescent powders is a common technique for externally marking adult mosquitoes [18,19,20,21]. This is to help assess multiple biological parameters, including population size, dispersal, mating and survival [22,23,24,25,26]. However, the interactions between the shipping/transportation, chilling and marking of sterile male mosquitoes has not yet been documented.

This study therefore aimed to develop and evaluate a mass transport technique for the long-distance shipment of sterile male mosquitoes from the laboratory to the field. In addition, simulated laboratory transport of marked and unmarked sterile males was assessed in terms of survival rates/recovery rates, flight ability and damage to the mosquitoes.

## 2. Materials and Methods

### 2.1. Biological Material and Rearing

For all experiments, the *Aedes aegypti* standard laboratory reference strains [27,28] were used. The *Aedes* strains were maintained following the “Guidelines for Routine Colony Maintenance of *Aedes* mosquitoes” [27]. An *Ae. aegypti* strain originating from Brazil (Juazeiro) was transferred to the IPCL from the insectary of Biofabrica Moscamed, Juazeiro, Brazil in 2012. In addition, a strain of *Ae. aegypti*, Dakar (Senegal strain) was transferred from the Laboratoire National de l’Elevage et de Recherches Vétérinaires, Institut Sénégalais de Recherches Agricoles, Dakar, Senegal, and was maintained in the same mass rearing conditions at the IPCL.

The rearing was conducted in controlled conditions of temperature (28 ± 2 °C), relative humidity (80 ± 10 RH%) and lighting (14:10 h light: dark, including 1 h of dawn and 1 h dusk for larval stages). Adult room conditions were 26 ± 2 °C, 60 ± 10 RH%, 14:10 h light: dark, including 1 h dawn and 1 h dusk.

To perform the experiments, mosquitoes were reared following modified mass rearing procedures developed at the IPCL [28,29,30]. Larval rearing started on Thursdays (day zero), when eggs were hatched and transferred to mass rearing trays. The trays were previously filled with 4 L of osmosis water on Fridays (day one) with 4% (*w*/*v*) of the IAEA reference diet. On day 7, pupae were collected once and sex-separated using mechanical and semi-automatic pupal sex-sorters (John W. Hock Co., Gainesville, FL, USA; Wolbaki, Guangzhou, China).

Plastic cups (600 mL) were used to aliquot 2000 male pupae and placed in cages (30 × 30 × 30 cm, BugDorm, BD4M1515, Taiwan) for emergence. Adult mosquitoes were maintained with access to a 10% sucrose solution until the day of the experiments.

### 2.2. Chilling, Compaction and Irradiation Procedures

On the day of irradiation, adult mosquitoes were transferred to the cold room at 4–5 °C to allow a knockdown for 10 min. Prior to the transfer, the sucrose solution was removed from the cages. Care was taken to ensure that no sugar or water were discarded into the cages. To prevent irregular mortality, no mosquitoes from wet cages were put into the batches to be transported.

After the knock-down, mosquitoes were removed from the cage by gently tapping the cages to drop the mosquitoes onto a paper towel that lined a 40 × 30 × 8 cm tray. Males were then transferred into individual mosquito compaction boxes (Diamond painting storage boxes) using a funnel in compacted batches of 100/cm^3^. Each individual compaction box lid was closed, and all of the boxes were covered with a mosquito net and rubber band to avoid any mosquitoes escaping through the small holes drilled into the lid. The mosquito compaction boxes were placed within a larger 11 × 17 cm containment box and were moved to the irradiation room using a transport box. The transport box was loaded with four ClimSel™ C7 packs to maintain a temperature between 4 and 8 °C. The transport box contained one ClimSel™ C7 pack at the bottom, which was covered with a towel, and two ClimSel™ C7 packs lined along the longest sides of the box. Two expanded pieces of polystyrene (EPS) (L × l × h: 5 × 5 × 1 cm) were also placed into the transport box to prevent direct contact between the mosquito boxes and the ClimSel™ C7 packs. The irradiation transport box was then closed and sealed using tape and carefully transported to the irradiation room.

Adult mosquitoes of both strains were exposed to a 45 Gy irradiation dose using an X-ray blood irradiator (Raycell MK2) [31]. Three 1 × 1 cm Gafchromic films were placed between the mosquito boxes prior to the irradiation to assess the actual dose received by the mosquitoes [32].

### 2.3. Mosquito Packing Procedures

After irradiation, sterile males were transported back to the cold room for packing using vacuum-insulated transport boxes (see Figure 1 and the Appendix A for the detailed protocol of the packing). In brief, two types of insulated boxes were used: a small carton thermobox (inner width × length × height = 17.5 × 17.5 × 17.5 cm) (Thermoboxes—RAJA (rajapack.at) and a bigger carton (CSafe Parcel R—5L—96 h) (inner width × length × height = 17.78 × 17.78 × 17.78 cm) (Parcel Solutions—CSafe passive parcel and small active solutions (csafeglobal.com)). The packing box had three levels of containment required for their transboundary transport to help avoid any possible escape [8].

### 2.4. Quality Control Parameters

For all experiments, the quality control parameters were survival (recovery rate after 24 h), flight ability and damage to the mosquitoes.

To determine the survival parameter, 100 chilled mosquitoes were collected using a mouth aspirator (about 1–1.5 cm within the aspirator’s tube of 0.8 cm diameter) [30] from a pile of chilled males in a cage (transported males). These samples were then transferred to 15 × 15 × 15 cm Bugdorm cages where males were allowed to recover and to feed on a 10% sugar solution for 24 h. All dead males were then removed from the cage using a mouth aspirator, while live mosquitoes were knocked down (−20 °C) and counted 24 h after unpacking. The level of survival (recovery rate) was calculated by dividing the number of live mosquitoes by the total number of males in each cage.

The flight ability was assessed using the routine device and protocol [30,33], immediately after unpacking the transport boxes (2 h) or after 24 h of recovery time.

To assess damage to mosquitoes such as missing legs and missing/damaged wings at the opening of the shipment, sterile male samples were taken immediately after unpacking from the pile of caged chilled mosquitoes. Samples were immediately stored in a fridge (4 °C) until the checking day on which dead males they are examined under a binocular stereoscope.

### 2.5. Environmental Conditions during Transport

For each shipment, two data loggers were placed in the transport box to record the temperature and relative humidity every ten minutes. These parameters were recorded for the simulated transport experiment (at the IPCL) and during the long-distance transport to Dakar, Senegal.

### 2.6. Mosquito Compaction Box Type, Recovery Time and Mosquito Quality during Long-Distance Transport

To assess the different effects of the compaction box type, two sets of boxes were studied: square boxes with and without drilled holes and different-sized round type boxes with v-shaped individual cells) (Appendix A).

For the square boxes, diamond painting boxes (2.5 × 2.3 × 2.3 cm, 13.3 cm^3^ volume) were used. Some had drilled holes and some were left without holes. Irradiated males of *Ae. aegypti*, Senegal strain, were then compacted (at a density of 100 mosquitoes/cm^3^) [30,34] and transported from the IPCL, Seibersdorf, Austria to Dakar, Senegal. The survival (recovery rate) and flight ability (capacity after recovery time following unpacking of 2 h and/or 24 h) of the transported males were assessed after delivery in Dakar. Two shipments were made with a 48 h transit time and one shipment was made with a 96 h transit time. The survival (recovery rate) and the flight ability parameters were compared between transport durations of 48 h and 96 h after 24 h post unpacking. Three to six replicates were assessed for each box type and for each of the three shipment events. In addition, the escape rate (flight ability) was assessed for the 48 h transit groups only to evaluate the effects of recovery time after unpacking (2 h and 24 h). Three to six replicates were assessed for each box type and for each of the two 48 h shipments.

Furthermore, the same parameters were assessed for the square boxes as compared to the round-type box with a v-shape for each of the eight individual compaction boxes (height 2.5 cm, 15 cm^3^ volume). Five and four replicates were assessed for each box type to assess survival (recovery rate) and flight ability (escape rate), respectively, from one shipment. The flight test was performed 24 h after unpacking.

All samples were taken from batches of 20,000 to 50,000 mosquitoes transported over a long-distance (*Ae. aegypti*, Senegal strain), from the IPCL, Seibersdorf, Austria (Europe), to Dakar, Senegal (Africa).

### 2.7. Long-Distance Transport, Irradiation and Mosquito Flight Ability

To assess the effects of 48 h and 96 h transport on mosquito quality, three shipments of *Ae. aegypti* Senegal strain (comprising between 20,000 and 50,000 sterile males each packed in square boxes with drilled holes), were sent from the IPCL, Seibersdorf, Austria, to Dakar, Senegal. The effects were assessed by comparing the escape rates of three groups: a control group of chilled, compacted but not irradiated or transported sterile males (‘control’); a group of chilled, compacted, irradiated but not transported sterile males (‘nontransported’) and a group of chilled, compacted, irradiated and transported sterile males that were sent to Dakar, Senegal (‘transported’). Four to six replicates were evaluated for each of the three shipments.

### 2.8. Marking, Irradiation, Transport and Mosquito Quality

To assess the effects of marking on sterile male mosquitoes during simulated transport under laboratory conditions at the IPCL, *Ae. aegypti*, Brazil strain, was used. Four transport boxes were prepared according to the mass transport protocol (Appendix A) and were stored at 4 ℃ the day before packing. Adults aged between three and four days were chilled for 10 min and transferred to the compaction boxes at a density of 100 males/cm^3^ until the day of opening, starting from 24 h to 96 h after packing. Mosquitoes were marked following the IPCL guidelines [35] using 10 mg of fluorescent dust for each batch of 2000 males, prior to the irradiation. After unpacking and sampling for damage and survival (recovery rate) checks, males were provided with a 10% sucrose solution and recovered for two hours prior to the flight test and were compared to untreated males (chilled, compacted but not irradiated males). The experiment was repeated three times with three to four replicates each. For each repeat, survival (recovery rates) after 24 h and flight ability (after 2 h and 24 h) were assessed. Damage to mosquito wings and legs was also assessed for two repeats. The experiments were performed under laboratory conditions of 26 ± 2 °C, 60 ± 10 RH% and a day-light regime (500–1000 lux).

The effects of marking, irradiation and transport were also assessed after 48 h of long-distance transport using, *Ae. aegypti*, Senegal strain. Two shipments of 36,000 mosquitoes were each made using round v-shaped transport boxes, both holding 6000 marked sterile males and 6000 unmarked sterile males. Samples for quality control were randomly taken from the individual mosquito compaction boxes containing about 1500 marked or unmarked males each. Four to six replicates were evaluated for each mosquito treatment including a control (irradiated and nonirradiated) and a marked and unmarked specimen from the groups of each shipment event.

## 3. Data Analysis

All statistical analyses were performed using R version 4.0.3 (https://cran.r-project.org, assessed on 1 June 2022) using RStudio (RStudio, Inc., Boston, MA, USA, 2016).

A generalized binomial linear mixed-effects model (GLMM) fit by maximum likelihood (Laplace approximation) with a logit link, with the escape rate (proportion of flyers) defined as the dependent variable whereby escaped (success or flyers) and nonescaped (failure or nonflyers) were weighted with the ‘cbind ()’ function and replicates defined as a random effect [36] to analyze the escape rate in each experiment.

For the effects of mosquito compaction box type, marking and duration of packing/transport on mosquito quality, box type (two levels: with and without holes or square and v-shaped boxes), marking (two levels: marked and unmarked), duration of packing/transport (four levels: 24 h, 48 h, 72 h and 96 h) and time after unpacking (two levels: 2 h and 24 h) were used as fixed effects and replicates were used as random effects in separate models.

For the morphological damage rate occurring during the simulated transport under laboratory conditions experiment, a binomial GLM was used, with the damage rate (proportion of damaged legs/wings) defined as the dependent variable whereby damaged (legs/wings) and nondamaged (legs/wings) were weighted with the ‘cbind ()’ function and with marking and duration of transport/packing as fixed effects.

The full models were checked for overdispersion (using Bolker’s function) [37] and for normality and homogeneity of variances on the residuals [38] for validation. When overdispersion in the model fit was detected, an observational column was added as a random effect. The new model fit was then considered for interpretation. To simplify the model, there was a stepwise removal of terms followed by likelihood ratio tests (LRTs). The minimal adequate model considered only factors that significantly reduced the explanatory power (*p* < 0.05) when removed [39]. The significant interactions were analyzed using the emmeans function (in package emmeans) [40]. All significant differences are based on *p* < 0.05.

### Ethical Statement

A permit N°01398/MEPA/DSV was granted to the Direction of Veterinary Services, Ministry of Livestock, to import sterile male *Ae. aegypti* (Senegal local strain) from the IAEA, Vienna, Austria, for research purposes including mosquito handling, transport and release to improve the trapping of sterile and wild males in the field.

## 4. Results

### 4.1. Environmental Conditions during Transport

Figure 2 shows the environmental conditions including temperature and relative humidity according to the transportation duration during long-distance transportation from Seibersdorf, Austria to Dakar, Senegal. Most of the shipment transit time lasted for 48 h (D3 after pick-up) where the mean temperature varied between 4 °C and 10 °C and up to 12 °C for 96 h. Relative humidity varied between 50% during packing to up to 60 to 70% after 96 h (post-pick-up).

During the simulated transportation experiments in the laboratory conditions at the IPCL Seibersdorf, Austria, temperature varied between 6 °C, 8 °C, 12 °C and 16 °C after 24 h, 46 h, 72 h and 96 h after packing, respectively. The relative humidity was around 70% regardless of the packing duration.

### 4.2. Mosquito Compaction Box Type and Recovery Time on Mosquito Quality during Long-Distance Transport

The best model considered the interaction between the type of mosquito box and the duration of transport to explain the decrease of mosquito escape rate (Table 1, χ^2^ = 41.235, df = 1, *p* = 0.0001, Figure 3).

The recovery time of mosquitoes which were transported for 48 h after unpacking showed a significant interaction between the type of mosquito compaction box and the time after unpacking (recovery time) (Table 2, χ^2^ = 13.7428, df = 1, *p* = 0.0002).

Overall, a 24 h recovery time increased male escape rates (holes: 0.86 (0.81–0.899, 95% CI); No_ holes: 0.712 (0.653–0.756, 95% CI)) regardless of the mosquito compaction box type as compared to a 2 h recovery time (0.562 (0.466–0.652, 95% CI)) for boxes with drilled holes or no drilled holes (0.665 (0.570–0.765, 95% CI)) (Figure 4).

When the effect of the type of square box was assessed in terms of mosquito survival (recovery rate) after long-distance transport, the best model included the interaction between the type of mosquito compaction box and the transit duration (Table 3, χ^2^ = 19.8158, df = 1, *p* < 0.0001).

After 48 h of transportation, more than 90% survival was recorded both in compaction boxes drilled with holes (0.94 (0.93–0.95, 95% CI)) and in compaction boxes without holes (0.91 (0.89–0.93, 95% CI)), which decreased to 50 and 70% (holes: 0.53 (0.46–0.61, 95% CI), No_ holes: 0.72 (0.65–0.78, 95% CI)) survival after 96 h (Figure 5).

Similar escape and survival rates were observed when two sizes of compaction box were used to transport mosquitoes (escape rate: square box (0.904 (0.87–0.93, 95% CI)), v-shaped box (0.888 (0.855–0.914, 95% CI)) and GLM: z value = −0.737, *p* = 0.461; survival rate: square box (0.974 (0.96–0.983, 95% CI)), v-shaped box (0.981 (0.966–0.990, 95% CI)) and GLM: z value = 0.919, *p* = 0.358)).

### 4.3. Long-Distance Transport and Irradiation on Mosquito Flight Ability

Long-distance transportation had a significant effect on male *Ae. aegypti*, Senegal strain, quality in terms of escape rate regardless of the duration of transport (48 h and 96 h) as compared to nontransported mosquitoes (Table 4, *p* < 0.0001). However, irradiation did not impact the mosquito escape rate (Figure 6, *p* = 0.936). A higher escape rate was observed when mosquitoes were transported for 48 h as compared to 96 h (Figure 6, z value = 4.720, *p* = 0.0001) with approximately a 20% reduction in the escape rate.

### 4.4. Marking, Irradiation, and Transport on Mass-Transported Sterile Male Mosquitoes

Simulating the mass transport of marked and unmarked sterile male *Ae. aegypti* mosquitoes, Brazil strain, in laboratory conditions showed that overall, marking (χ^2^ = 10.29, df = 1, *p* = 0.001341, Figure 7), packing duration (24 h, 48 h, 72 h and 96 h) (χ^2^ = 318.99, df = 6, *p* < 0.0001, Figure 7) and time after unpacking (2 h or 24 h-recovery times) (χ^2^ = 115.38, df = 1, *p* < 0.0001, Figure 7) had an impact on mosquito quality in terms of the escape rate (Appendix A).

The packing of *Ae. aegypti*, Brazil strain, for a duration from 24 h up to 96 h reduced the mosquito survival rate (recovery rate) after 24 h following unpacking (Table 5, Figure 8, χ^2^ = 608.84, df = 3, *p* < 0.0001). Marking also had a negative effect (Table 5, Figure 8, χ^2^ = 6.34, df = 1, *p* = 0.01181).

Marking did not damage the legs of the packed *Ae. aegypti*, Brazil strain, for a duration from 24 h up to 96 h (χ^2^ = 0.3401, df = 1, *p* = 0.55979) unlike the packing duration (χ^2^ = 11.9527, df = 4, *p* = 0.01771). Marking also had no negative effect on mosquito wings.

When the effects of marking, irradiation and transport were assessed after transport with a two-day duration were investigated using *Ae. aegypti,* Senegal strain, marking had no negative impact on the mosquito escape rate (Figure 9, χ^2^ = 0.308, df = 1, *p* = 0.579), unlike the long-distance transport itself (Figure 9, χ^2^ = 41.235, df = 1, *p* < 0.0001). Irradiation did not reduce the mosquito quality in terms of the escape rate (Figure 9, χ^2^ = 37.839, df = 1, *p* = 0.4294).

Marking had no effect on the survival (recovery rate) of the adult *Ae. aegypti*, Senegal strain as compared to unmarked sterile males (marked males: (0.92 (0.881–0.947, 95% CI)); unmarked males: (0.91 (0.87–0.939, 95% CI)), GLM: t = −0.421, *p* = 0.682).

## 5. Discussion

Our study showed that it is possible to transport chilled sterile males for a period of two to four days over a long distance using two types of mosquito compaction boxes. In addition, it has been shown that irradiated males can be marked prior to shipment but with a moderate cost in terms of recovery rate (survival) and male quality as compared to nonmarked sterile males. A recovery time of 24 h improved male quality as compared to a recovery time of 2 h.

The SIT is based on the release of sterile male insects that should be competitive against their wild population counterparts for attracting females. Males are produced simultaneously and in high numbers to ascertain that a program is operational for a sufficient time to observe an impact on the targeted population. The success of the technique relies on a high survival rate and quality of released sterile males, which also help in reducing the production cost. The cost reduction may also rely on the use of optimal tools and protocols for mass production [29,41,42], optimal irradiators and irradiation doses [31], quality control [30,33,43] and also the release strategy in the field [24,25,26]. Most published studies relating to field work rely on irradiated mosquitoes that were produced and delivered at field sites either as pupae that need time to emerge and are kept for a small number of days prior to release or as adults ready to be released. For instance, pilot release studies in Germany [14], Montenegro and Greece [44] and Albania [24] relied on outsourced sterile male mosquitoes that were sent from Italy. A method of transport for nonchilled adults was used, but chilled adults were also transported for a period not exceeding 24 h. However, most of the transboundary shipments by air for longer distances will take more than 24 h considering the time it takes for the shipment to be picked up and delivered. Therefore, developing shipping containers/boxes that both work well and require less space requirement will increase the cost-efficiency of SIT programs. We observed here that there was an interaction between the type of mosquito box and the duration of transport on the mosquito escape rate. This shows that while chilling reduces the required space and reduces the cost during long-distance transport, its effects may vary depending on the type of mosquito compaction box and the duration of transport. Holes drilled in the tsetse transport boxes allowed ventilation, stable temperature (mean ± sd: 10.1 ± 2.3 °C) and relative humidity (mean ± sd: 81.4 ± 8.7%) conditions during shipment [45,46,47]. These authors also described that irradiated pupae were placed in Petri dishes and packed in insulated boxes during long-distance transportation from Bobo-Dioulasso, Burkina Faso, to Dakar, Senegal, to assess the quality of the delivered flies. When mosquitoes were packed for a longer period (96 h) in compaction boxes without drilled holes, there was a trend of an increased escape rate as compared to mosquitoes packed in boxes with holes which suggests a protective effect of anoxia. Anoxia also led to a greater adult flight ability in irradiation-stressed flies [48]. Yamada and collaborators have also recently found that chilling induces damage to mosquitoes when treated as adults, reducing longevity and flight ability, but partial or full recovery is possible if chilling duration and temperature are carefully controlled [49]. In addition, we found in our study that the increase in temperature had probably awoken the mosquitoes and led to an increase in mortality. However, further studies may be needed to assess different atmospheric conditions during long-distance mass transportation of irradiated male mosquitoes.

The protocol developed here has shown that transported males could survive in good numbers (over 90% after two days and between 70% and 50% after four days) when mosquitoes were shipped at a density of 100 males/cm^3^ regardless of the type or the size of the box. A similar survival rate of 85% was previously observed by Chung et al. [50] when *Ae. aegypti* mosquitoes were compacted at a density of 240 individuals/cm^3^ and shipped via air freight overnight from New Mexico to California. In contrast, Mastronikolos et al. [9] found that shipment from Italy to Greece led to approximately 15% and 40% mortality for 24 h total transit time and nearly 48 h total transit time, respectively.

While males may be released immediately after delivery, it was shown that a recovery time of 2 h may not be enough to allow the majority of sterile males to regain their flight capacity. Several factors, including low temperature, shocks and pressure may have impacted their flight capacity. It is known that long exposure to cold [10,30,51] can reduce mosquito quality. Several authors have observed this phenomenon (reviewed in [52]). A temperature range between 7 and 12 °C could be tolerated for shipping chilled *Ae. aegypti* mosquitoes for a period of up to 72 h without creating an adverse mortality rate. A 24 h recovery time enhanced male quality, which suggests that males transported over a long-distance should be allowed to recover for at least this length of time whenever possible and cost-effective. It has been shown that the adverse effect of chilling on the *Ae. aegypti* mosquito population escape rate was undetected after 24 h of recovery [34]. Oliva et al. [53] have previously shown that the time spent in the insectarium during the pre-release period could have increased *Ae. albopictus* (La Reunion island strain) sugar reserves and thus improved survival and flight capacity. Given the low capacity of male *Aedes* mosquitoes to disperse in the field, there is a renewed interest in areal releases using drones [17,54]. It is easier to load the drone release device with chilled mosquitoes as soon as they are delivered. However, a second short chilling may be needed after a recovery period prior to loading the drone to minimize the impact on the quality, but further study is required to prevent the potential negative impact and to evaluate the cost-effectiveness of these two strategies.

However, we found a significant impact of fluorescent marking on the mosquito escape rate when packing and transport were simulated on the *Ae. aegypti*, Brazil strain. Indeed, an above 70% escape rate was observed even for 48 h of transport-simulated packing when mosquitoes were allowed to recover for 24 h. Marked and unmarked mosquitoes maintained high escape rates (>75%) after 2 h only of recovery time and even more after 24 h of recovery time. In addition, packed irradiated male *Ae. aegypti* survived well (>85%) given 72 h of packing. These results showed for the first time that chilled sterile male *Ae. aegypti* mosquitoes can be marked prior to a long-distance shipment from Europe to Africa.

## 6. Conclusions

This study demonstrated that the developed protocol for the long-distance transport of chilled, irradiated and marked adult mosquitoes may be a useful tool for SIT pilot programs worldwide. A single transport box can hold up to 50,000 sterile male mosquitoes and cost about EUR 230 (including the cost of a single box and transportation fees) to be sent from Austria to Africa by an air freight courier. Therefore, alternative materials to lower the cost of the long-distance transport system could also be investigated for developing countries. Although promising results were achieved, further studies are required to assess the maximum capacity of sterile males to be transported per transport box and to assess the effects of long-distance transport on irradiated male survival, mating competitiveness and dispersal in the field.

## Figures and Tables

**Figure 1 insects-14-00207-f001:**
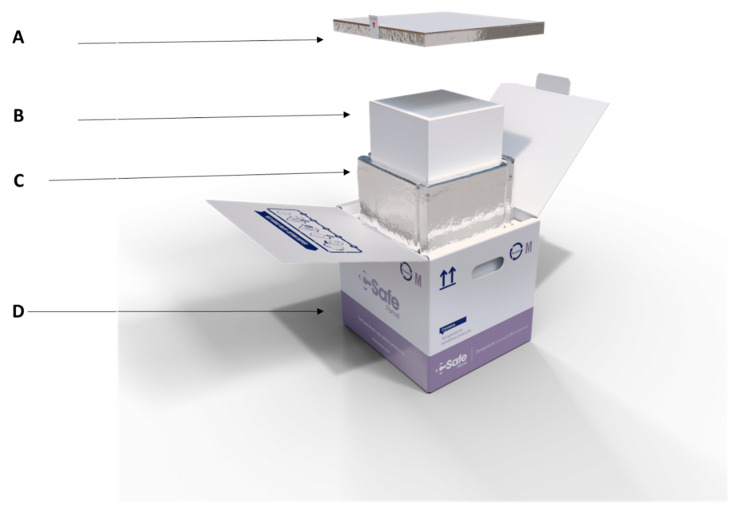
Assembly of the packing box (mass transport box) including the lid (**A**), the inner carton (small carton) (**B**), the outer carton (big carton vacuum insulated box) (**C**) and an external protection (**D**) (csafeglobal.com).

**Figure 2 insects-14-00207-f002:**
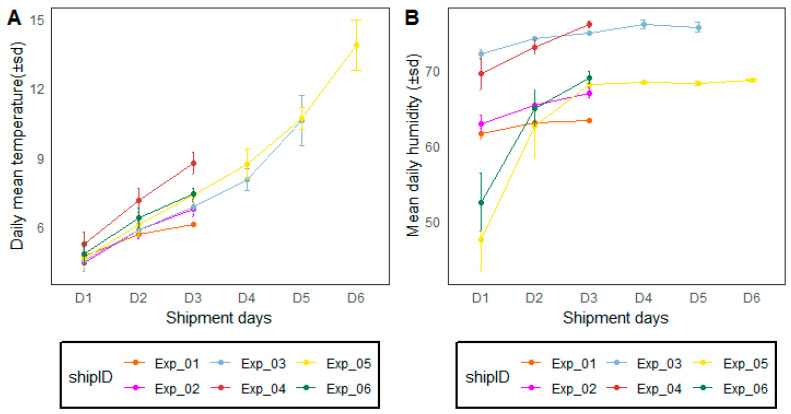
Environmental condition variations including temperature (**A**) and relative humidity (**B**) of six shipments from Seibersdorf, Austria, to Dakar, Senegal, during long-distance transport experiments. Vertical lines separate dates. ‘ShipID’ stands for ‘shipment number (Exp = experiment 01 to 06). ‘D’ stands for ‘shipment day’. Data were recorded using loggers placed at the bottom and top of the mosquito compaction boxes. Loggers were in contact with the Phase Change materials.

**Figure 3 insects-14-00207-f003:**
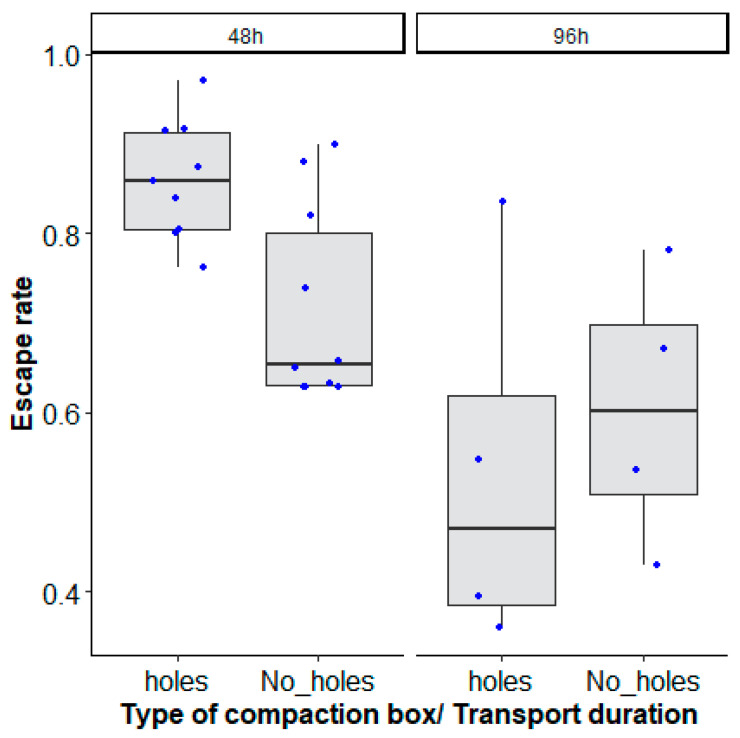
*Aedes aegypti*, Senegal strain, escape rates according to the type of mosquito compaction box: square with holes (‘holes’) and without holes (‘No_ holes’) and duration of transport (48 h and 96 h). Black bars indicate the median. The upper and lower limits of each box indicate the interquartile range. Each blue dot represents a value of the observed escape rate per replicate.

**Figure 4 insects-14-00207-f004:**
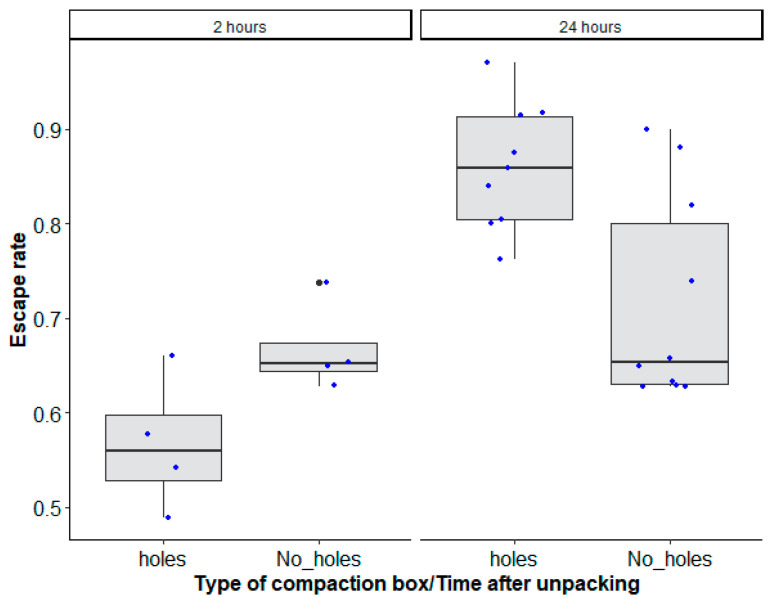
*Aedes aegypti*, Senegal strain, escape rates according to the type of mosquito compaction square box with holes (‘holes’) and without holes (‘No_ holes’), and time after unpacking (2 h and 24 h recovery times). Black bars indicate the median. The upper and lower limits of each box indicate the interquartile range. Each blue dot represents a value of the observed escape rate per replicate. Black dots represent outliers.

**Figure 5 insects-14-00207-f005:**
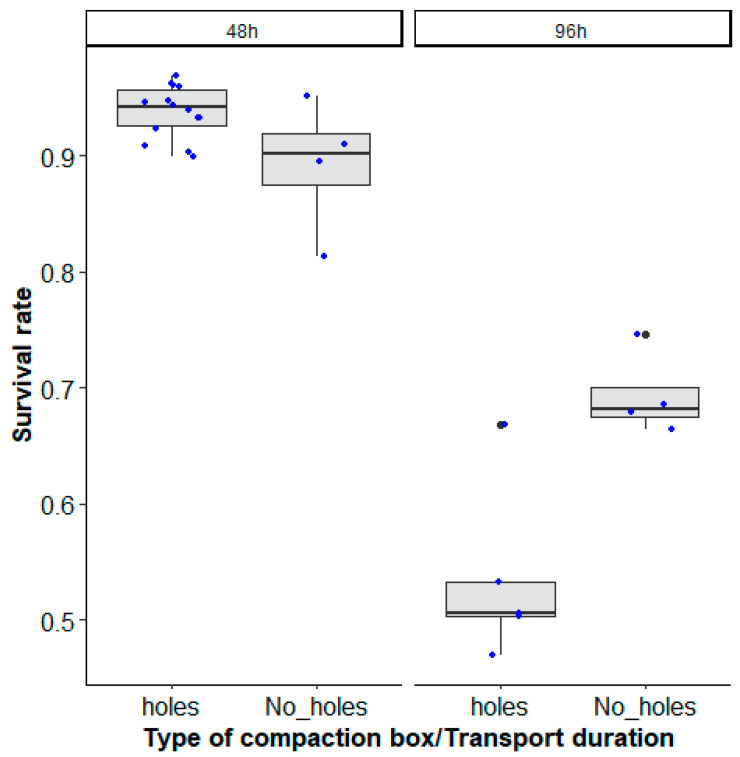
*Aedes aegypti*, Senegal strain, survival rates (recovery rates) according to the type of mosquito compaction square box with holes (‘holes’) and without holes (‘No_ holes’) and the duration of transport (48 h and 96 h). Black bars indicate the median. The upper and lower limits of each box indicate the interquartile range. Each blue dot represents a value of the observed escape rate per replicate. Black dots represent outliers.

**Figure 6 insects-14-00207-f006:**
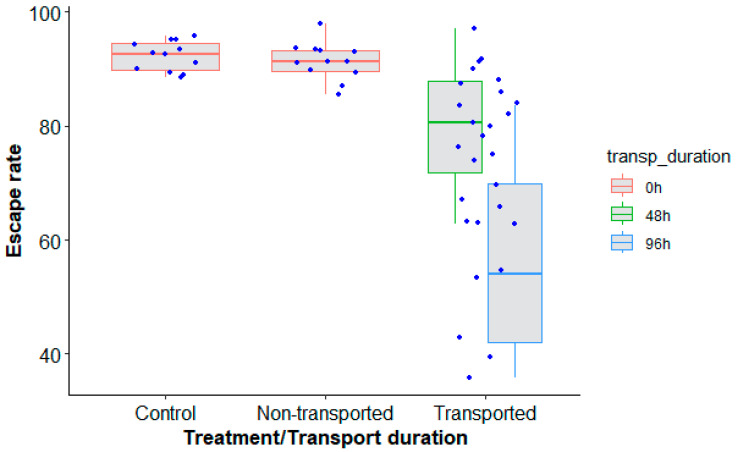
*Aedes aegypti*, Senegal strain, escape rates according to the type of mosquito treatment (control, transported and nontransported) and duration of transport (0 h (control and nontransported), 48 h and 96 h). Black bars indicate the median. The upper and lower limits of each box indicate the interquartile range. Each blue dot represents a value of the observed escape rate per replicate.

**Figure 7 insects-14-00207-f007:**
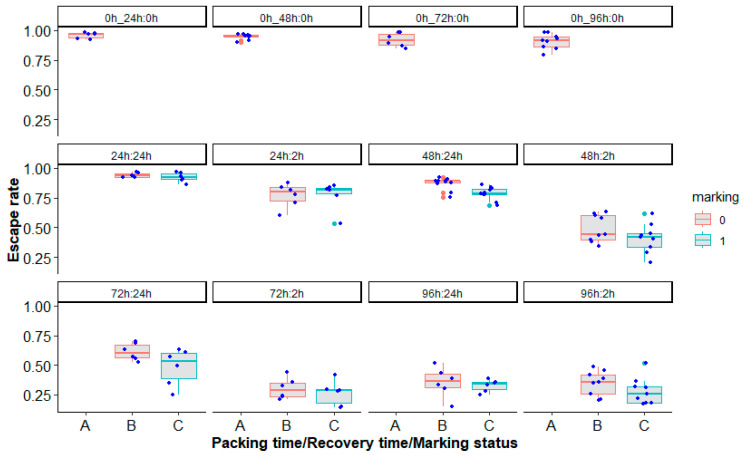
Escape rates according to the marking status (A = control, B or 0 = unmarked, C or 1 = marked), packing duration (0 h, 24 h, 48 h, 72 h and 96 h) and the time after unpacking (0 h, 2 h and 24 h recovery times) of *Aedes aegypti*, Brazil strain, in laboratory conditions. Control groups were assessed daily for packing time and were labeled as 0 h_24 h:0 h, 0 h_48 h:0 h, 0 h_72 h:0 h or 0 h_96 h:0 h panels. Blue and red bars indicate the median. The upper and lower limits of each box indicate the interquartile range. Each blue dot represents a value of the observed escape rate per replicate. Green and red dots represent outliers.

**Figure 8 insects-14-00207-f008:**
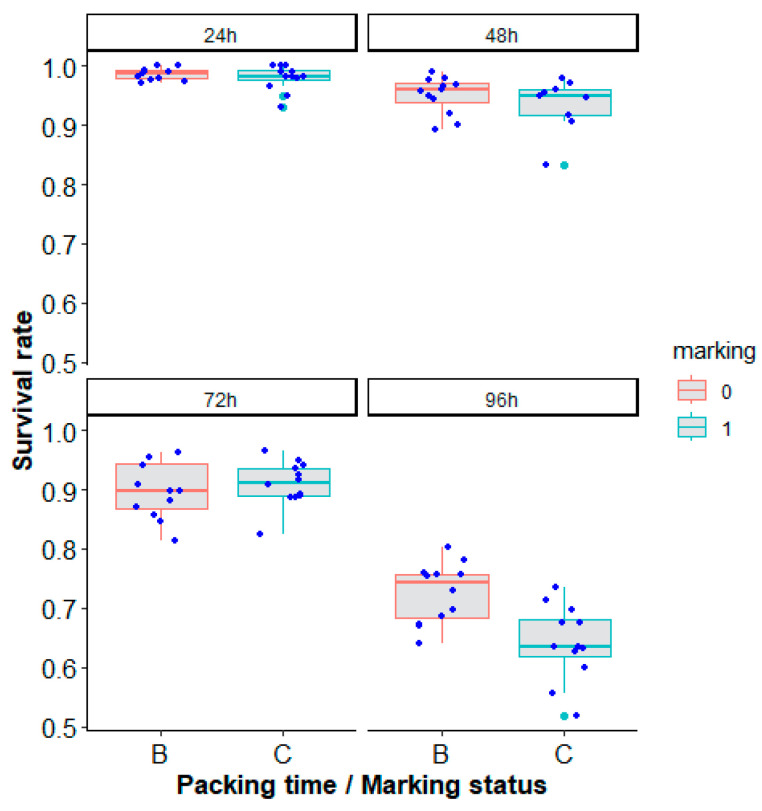
Survival rate (recovery rate) according to marking status (B or 0 = unmarked, C or 1 = marked) and packing duration (24 h, 48 h, 72 h and 96 h panels) of *Aedes aegypti*, Brazil strain, in laboratory conditions. Each blue dot represents a value of the observed survival rate per replicate. Green dots represent outliers.

**Figure 9 insects-14-00207-f009:**
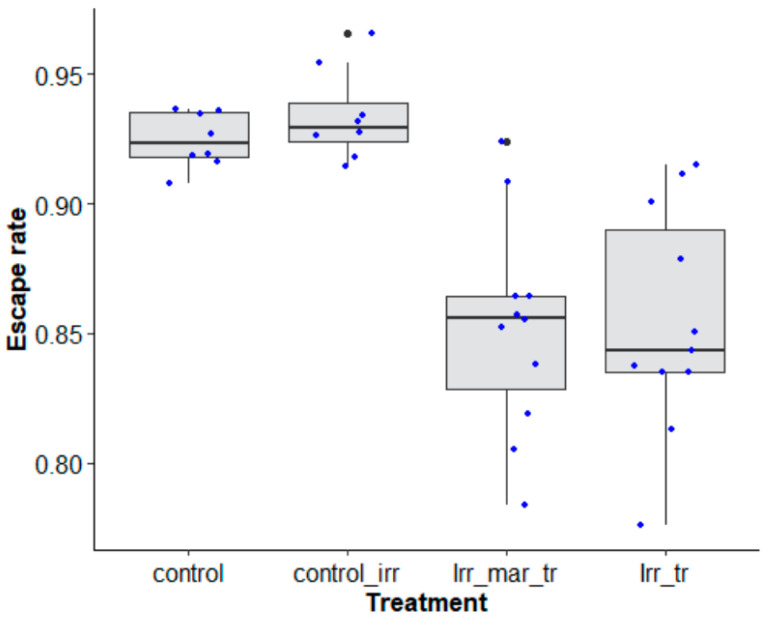
Escape rates according to mosquito treatment (control, marked and unmarked) of *Aedes aegypti*, Senegal strain, transported over a long distance. Control groups (control and control_ irr = control irradiated) were assessed at the laboratory of origin (Austria) while other treatments (Irr_ mar_ tr = irradiated_ marked- transported and Irr_ tr = irradiated_ unmarked-transported) were assessed at the receiving laboratory (Senegal). Black bars indicate the median. The upper and lower limits of each box indicate the interquartile range. Each blue dot represents a value of the observed escape rate per replicate. Black dots represent outliers.

**Table 1 insects-14-00207-t001:** Fixed effects of mosquito compaction box type and recovery time on the escape rate of *Aedes aegypti*, Senegal strain, during long-distance transport.

	Estimate	Std. Error	z Value	Pr(>|z|)	
(Intercept)	1.8157	0.1616	11.236	<2 × 10^−16^	***
Box type: No_holes	−0.8894	0.1151	−7.731	1.07 × 10^−14^	***
Transport duration: 96 h	−1.6499	0.2842	−5.806	6.41 × 10^−09^	***
Box type No_holes: Transport duration 96 h	1.1987	0.1867	6.421	1.35 × 10^−10^	***

Signif. codes: ‘***’ 0.001.

**Table 2 insects-14-00207-t002:** Fixed effects of the interaction between mosquito compaction box type (square with and without holes) and recovery time (2 h and 24 h) after unpacking on *Aedes aegypti*, Senegal strain, escape rate after long-distance transport.

	Estimate	Std. Error	t Value	Pr(>|t|)	
(Intercept)	0.2475	0.1951	1.268	0.21744	
Box type: No_holes	0.4391	0.2841	1.545	0.1359	
Time after unpacking: 24 h	1.5707	0.2709	5.799	6.58 × 10^−6^	***
Box type No_holes: Time after unpacking 24 h	−1.35	0.368	−3.668	0.00128	**

Signif. codes: ‘***’ 0.001; ‘**’ 0.01.

**Table 3 insects-14-00207-t003:** Fixed effects of the interaction between mosquito compaction box type (square with and without holes) and the duration of transport on *Aedes aegypti*, Senegal strain, survival rate (recovery rate) after long-distance transport.

	Estimate	Std. Error	z Value	Pr(>|z|)	
(Intercept)	2.7557	0.1033	26.681	<2 × 10^−16^	***
Box type: No_holes	−0.458	0.2024	−2.263	0.0236	*
Transport duration: 96 h	−2.6093	0.1901	−13.723	<2 × 10^−16^	***
Box type No_holes: Transport duration 96 h	1.1862	0.2665	4.451	8.53 × 10^−6^	***

Signif. codes: ‘***’ 0.001; ‘*’ 0.05.

**Table 4 insects-14-00207-t004:** Fixed effects of mosquito treatments (transported and nontransported) and the duration of transport (48 h and 96 h) on *Aedes aegypti*, Senegal strain, escape rate after long-distance transport.

	Estimate	Std. Error	t Value	Pr(>|t|)	
(Intercept)	2.45413	0.23472	10.456	7.45 × 10^−14^	***
Nontransported	−0.02639	0.32512	−0.081	0.936	
Transported	−2.19028	0.29182	−7.506	1.41 × 10^−9^	***
Transport duration: 48 h	1.02543	0.21725	4.72	2.15 × 10^−5^	***

Signif. codes: ‘***’ 0.001.

**Table 5 insects-14-00207-t005:** Fixed effects of the effects of marking (1 = marked vs. 0 = unmarked), irradiation and transport duration (packing times: 48 h, 72 h and 96 h vs. 24 h) on the escape rate of *Aedes aegypti*, Brazil strain, in laboratory conditions.

	Estimate	Std. Error	t Value	Pr(>|t|)	
(Intercept)	4.11789	0.20141	20.446	<2 × 10^−16^	***
Marking: 1	−0.23563	0.09365	−2.516	0.0137	*
Packing time: 48 h	−1.11229	0.23399	−4.754	7.74 × 10^−06^	***
Packing time: 72 h	−1.71948	0.21787	−7.892	7.68 × 10^−12^	***
Packing time: 96 h	−3.20626	0.20431	−15.693	<2 × 10^−16^	***

Signif. codes: ‘***’ 0.001; ‘*’ 0.05.

## Data Availability

Data is contained within the article or supplementary material. The data presented in this study are available in Appendix A.

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
