# Peer review of "From the Lab to the Field: Long-Distance Transport of Sterile Aedes Mosquitoes"

_insects, 2023, doi:10.3390/insects14020207_

Round 1

Reviewer 1 Report

Maiga and collegues present a very interesting study on the long-distance transportation of Aedes sterile males for Sterile Insect Technique programs.

The paper is very well written and results are clearly presented. In addition, a detailed protocol is provided to readers as supplementary material, representing a very useful resource to adopt the outcomes of the present paper in other tests/studies.

Hence, with very minor revisions (mainly text typos), I strongly agree with the publication of this MS in your journal.

Minor revision

Please, change the species names in Italics in the whole paper.

Please, check for double spaces in the whole text (example: line 79, between for and mosquitoes words).

Please, clarify the text at line 273.

Please, correct Y-axis label of fig. 2A (Temperature in degree celsiu) and provide a caption for figure 2 with a clear explanation of the three figure section (A, B and C). Please add letter C to the figure.

Please, re-format the figure style in order to have a unique set for all the figures. Label font size and style are differet from figure to figure. 

Author Response

Minor revision

 Please, change the species names in Italics in the whole paper.

Answer: Species names have been italicized as suggested. Thank you.

 Please, check for double spaces in the whole text (example: line 79, between for and mosquitoes words).

Answer: the text was checked accordingly. Thank you

 Please, clarify the text at line 273.

Answer: the sentence has been shorten to avoid confusion

 Please, correct Y-axis label of fig. 2A (Temperature in degree celsiu) and provide a caption for figure 2 with a clear explanation of the three figure section (A, B and C). Please add letter C to the figure.

Please, re-format the figure style in order to have a unique set for all the figures. Label font size and style are differet from figure to figure.

 Answer: Figure has been changed with a better view and caption. Thank you. 

Reviewer 2 Report

The document is well written, it contributes to the knowledge about the long-distance transport of sterile Aedes mosquitoes. The introduction is clear and precise, only in line 74 complete the parentheses. The materials and methods are well-written. Please complete, For damages, …………………... The results section needs more description, it is very accurate to describe the models, but some comments should be added about what the result indicates. Clarifications are required, which are described in the pdf document. In the discussion describe the conditions that improve long-distance transport.

Author Response

Reviewer 2

All the revision has been highlighted in yellow in the manuscript for reviewer 2.

Line 49. Abstract: morphological damage? to be more specific

Answer: ‘morphological’ has been added to ‘damage’ as suggested

Line 50. ‘with a non-significant. Certainly the damage is not significant but there is a significant decrease after 4 days. Highlight what is related to the fact that the quality is maintained for 48h, enough to transport

Answer: ‘with a non-significant’ has been added as suggested. Thank you.

Line 74. Complete the parenthesis

Answer: The parenthesis has been completed as suggested. Thank you.

Line 290. Results need to be accurately described

Answer: The results have been now described. Figure has been changed with a better view and caption. Thank you.

Line 300. Briefly describe.

Answer: The Figure 2 has been described as following ‘Figure 2 shows the environmental conditions including temperature and relative humidity according to the transportation duration during long-distance transportation from Seibersdorf, Austria to Dakar, Senegal. Most of the shipment transit time lasted for 48h (D3 after pick-up) where the mean temperature varied between 4 °C and 10 °C and up to 12 °C for 96h. Relative humidity varied between 50% during packing to up to 60 to 70% after 96h (post pick-up).’

Line 301. 4.2. Title. Describe the quality parameters

Answer: Quality parameters have been described at the material and Methods section above namely ‘2.4.Quality control parameters’. We did not want to lengthen the title but adding the details. Thank you

Line 304. The effect was to increase or decrease or no change was observed

Answer:  The effect was decreased. The sentence has been rewording as following ‘ The best model considered the interaction between the type of mosquito box, and the duration of transport to explain the decrease of mosquito escape rate (Table 1, χ2= 41.235, df= 1, p = 0.0001). ‘

Line 346. Add ‘decrease to’ before ‘50’

Answer: ‘decrease to’ has been added before ‘50’ as suggested. Thank you

Lines 450-457. ‘Males are produced simultaneously and in high numbers to ascertain that an operation-al programme is running operational for a sufficient time to see impact on the targeted population. The success of the technique relies on a high survival rate and quality of re-leased sterile males, which also help reducing the production cost. The cost reduction may also rely on the use of optimal tools and protocols for mass production [29, 41-42], optimal irradiators and irradiation doses [31], quality control [33, 30, 43] but also release strategy in the field [24-26]’. Doesn’t really contribute to the discussion.

Answer: This paragraph was a reminder of SIT pckage which mentioned parameters such as survival and other quality control parameters. We felt like it was good to share this reminder here and follow up with transport studies. We wish with respect to keep the relevant references here. Thank you

Lines 472. ‘Holes drilled in the tsetse transport boxes allowed ventilation and stable temperature and relative humidity conditions during shipment [45-47]’. ‘Describe these conditions’

Answer: .The conditions have been described now as following in the manuscript ‘Holes drilled in the tsetse transport boxes allowed ventilation and stable temperature (mean ± sd: 10.1 ± 2.3 °C) and relative humidity (mean ± sd: 81.4 ± 8.7%) conditions during shipment [45-47]. ‘

Line 513. Replace ‘understand’ by ‘avoid/prevent’

Answer: ‘understand’ has been replaced by ‘prevent’ as suggested.

Additional comments from the Editor

Hope this email finds you well.

We sent you your manuscript for minor revision recently.

There are some additional points that need to be included:

1. Please reduce the overlaps marked in red (in the main text) in the
file attached to this email (refers to the paper "Standardization of the
FAO/IAEA Flight Test for Quality Control of Sterile Mosquitoes");

The overlaps have been reduced wherever possible. There are unavoidable overlaps such as laboratory names, material and methods titles, statistical analysis, figure caption…. We tried to reduce the overlaps as much as possible for your consideration. The sentences have been highlighted in blue.

  1. Self-citation rate should be limited to 15%. We understand that
    relevant references are cited and important for the work. The current
    rate is 31%, you can remove several or add more relevant references to
    fit in 15%;

Answer: There are in total 7 references with Maiga (or Maïga) among the authors (6/54= 11.1%) from which 2 as first author (2/54= 3.7%) if we are not wrong. We probably did not understand your calculations, but we will be happy to reduce the self-citation to fit the 15% should it apply to our manuscript. Thank you

  1. Please pay attention to the back matter of the paper (Data
    Availability Statement for the type Article cannot be "Not applicable");
    Answer: the data have been now included in the supplementary files.
